# A Post-Hoc Analysis of Depressive Disorders in Patients with Type 2 Diabetes

**DOI:** 10.3390/healthcare13151773

**Published:** 2025-07-22

**Authors:** Yegan Pillay, W. Guyton Hornsby, Chandan K. Saha, Jay Shubrook, Kent A. Crick, Ziyi Yang, Kieren Mather, Mary de Groot

**Affiliations:** 1Department of Counseling and Higher Education, Ohio University, Athens, OH 45701, USA; pillay@ohio.edu; 2Department of Exercise Physiology, West Virginia University, Morgantown, WV 26506, USA; ghornsby@hsc.wvu.edu; 3Indiana University School of Medicine, Indiana University, Indianapolis, IN 46205, USA; cksaha@iu.edu (C.K.S.); ziyiyang@iu.edu (Z.Y.); 4Department of Primary Care, Touro University California, Vallejo, CA 94592, USA; jay.shubrook@tu.edu; 5Counseling and Psychological Services, University of California Los Angeles, Los Angeles, CA 90095, USA; kcrick13@gmail.com; 6Eli Lilly & Co., Indianapolis, IN 46285, USA; kierenmather@gmail.com

**Keywords:** diabetes, depression, survival analysis

## Abstract

**Background/Objectives:** This study is an investigation of the occurrence, remission and recurrence of major depressive disorders (MDDs) in adults with type 2 diabetes (T2DM). **Methods:** Interviews were conducted with individuals (N = 176) who met the criteria for MDD using the Structured Clinical Interview for the DSM-IV-TR (SCID). **Results:** N = 176 T2DM adults, with a mean (SD) age of 55.5 (10.4) years, 74% of whom were female and 62% were white, completed the Structured Clinical Interview for the DSM-IV-TR (SCID). A mean (SD) number of 1.8 (0.9) episodes of major depression (MDD) were recorded from birth to the date of interview, with a mean (SD) onset age of the first episode of 40.4 (15.9) years. Median (IQR) MDD episode duration was 13.9 (5.6–31.9) months and the median (IQR) cumulative lifetime exposure duration to MDD was 33 (12.9–63.1) months. Kaplan–Meier survival analysis along with the frailty model, to account for the correlation among multiple recurrences or remissions within a subject, indicated that the median first episode duration was shorter than the median second episode duration (14 vs. 37.9 months, *p* < 0.0001). Of those who had at least three episodes, the median second episode duration was shorter than the median third-episode duration (13.0 vs. 28.0 months, *p* = 0.006). The median recurrence time following first remission was significantly longer than the median recurrence time following second remission (138.0 vs. 80.6 months, *p* = 0.02). **Conclusions:** These results document that clinical depression is recurrent in adults with T2DM. Moreover, depressive episodes in individuals with T2DM are persistent well beyond episode durations observed in the general population.

## 1. Introduction

The World Health Organization has documented that 5% of adults and 5.7% of older adults 60 years or older experience depression, representing more than 280 million people worldwide. The prevalence of depression increased by 25% during the COVID-19 pandemic [1]. In the United States, the 2023 National Survey on Drug Use and Health (NSDUH) reported that approximately 21.9 million (8.5%) of adults aged 18 years or older experienced a major depressive episode (MDD), with prevalence rates for adults aged 18–25 equaling approximately 6 million (17.5%) followed by adults aged 26–29 accounting for 10.6 million (10.2%) and those 50 years or older representing approximately 5.4 million (4.5%) [2]. Adult females (10.3%) reported a higher incidence of major depressive episodes than males (6.2%) and those who reported having more than two races had the highest prevalence rate of 13.9% [3,4,5].

Worldwide, diabetes affects more than 830 million adults, with more than 90% experiencing type 2 diabetes [6]. In the United States, type 2 diabetes (T2DM) affects nearly 38.3 million adults, with another 98 million affected by pre-diabetes. In addition to premature mortality, T2DM is a significant contributor to overall health care expenditures. In 2022 alone, T2DM accounted for USD 413 billion of total medical expenditure, of which more than USD 90 million of this cost was due to reduced productivity as a result of absenteeism, unemployment due to chronic disability, and premature mortality [7].

Diabetes and depression have high rates of comorbidity [8,9,10]. A meta-analysis of 15 studies conducted prior to 2000 reported prevalence rates of elevated depressive symptoms to be 27% in people with T2DM [9]. Another meta-analysis of 10 studies conducted with a total of 51,331 people across several countries reported an overall depression prevalence of 17% [10]. A study using the US National Health and Nutrition Examination Survey (NHANES) 2005–2012 data reported a lower point prevalence rate of depression (10.6%) [11]. However, when antidepressant medication and clinically relevant depression were combined, it raised the burden of depression on people with T2DM to 25.4% overall and 35% in women [11]. There is substantial evidence that comorbid depression and T2DM is significantly associated with worsened glycemic control [12], increased severity of diabetes complications [13], poorer adherence to diabetes self-care [14,15], decreased quality of life [16,17], greater functional disability [18] and early mortality [19].

The number of studies that have examined the characteristics of depression using standardized psychiatric interview protocols to distinguish depressive disorders from symptoms otherwise attributable to T2DM or other medical comorbidities has been limited. The results of a 10-year follow-up study by Coleman and his colleagues in a sample of 4128 used an ICD-10 algorithm combined with death certificate data to classify mortality types among patients with type 2 diabetes found a significant positive relationship between depression and mortality with major depression, demonstrating a stronger relationship than minor depression [19].

Our research team previously examined the duration of MDD and other mood disorders in a sample of 50 adults with T2DM and MDD [20]. In this sample, the average number of MDD episodes was 1.8, with an average MDD episode lasting 23.4 months (SD 31.9, range 0.5–231.3). When all depressive diagnoses were combined, the total lifetime exposure to depression was 43.1 months (SD 46.5, range 0.5–231.3).

It is evident, based on the previous studies, that clinical depression is recurrent in both the general population and in samples that were drawn from T2DM patients. A significant limitation of the studies that have assessed individuals with diabetes using psychiatric interviews is the lack of generalizability due to the relatively small sample sizes and the homogeneity of the groups. To address this limitation, the current study was designed to investigate the lifetime course of MDD and other depressive disorders in terms of episode length, remission duration, and progression of episodes in a sample of adults with T2DM drawn from rural and urban communities. In addition, we examined MDD episode and remission duration before and after the diagnosis of T2DM to test whether the order of diagnosis influenced the burden of depression.

## 2. Materials and Methods

The current study represents a secondary data analysis of the Program ACTIVE II clinical trial [21,22,23,24,25]. A four-arm randomized controlled comparative effectiveness study was conducted during 2011–2016 to test combination and individual behavioral approaches to treating depression in adults with T2DM against usual care. Participants were recruited primarily from physician practices and through the utilization of various forms of community advertisements in southeastern Appalachian Ohio, West Virginia, and central Indiana. The study was approved by the Ohio University, West Virginia University, and Indiana University Institutional Review Boards. All participants provided written informed consent after receiving a complete description of the study and study procedures. Eligibility for participation in the study required being 18 years of age, a diagnosis of current major depressive disorder based on DSM-IV-TR diagnostic criteria, and being ambulatory. Participants were ineligible if they had severe valvular heart disease or had a cardiac event in the past year, class III or IV heart failure, atrial fibrillation, uncontrolled stage 2 hypertension, laser surgery to treat diabetic retinopathy in the previous 6 months, a history of stroke, lower limb amputation, asensory peripheral neuropathy, severe chronic obstructive pulmonary disease, or were medically unstable. Participants with a lifetime history of bipolar depression, current suicidal ideation, substance abuse or dependence, or psychotic symptoms were also excluded. A detailed description of the methods used in the primary Program ACTIVE II study are described in detail elsewhere [22].

For the current study, participants were assessed at the point of study enrollment. A total sample of N = 176 completed psychiatric interviews. The current study captured lifetime history and course of depression from birth to study enrollment.

### 2.1. Measures

*Demographic Data.* Demographic characteristics were obtained via a self-administered questionnaire completed by each participant at baseline. Variables included the following: age, ethnicity, marital status, income, educational status, work status, and health insurance status.

Assessment of Depression: Structured Clinical Interview for the Diagnostic and Statistical Manual of Mental Disorders-IV-TR (SCID). The SCID is a semi-structured psychiatric diagnostic interview protocol that assesses current and lifetime presence of 33 Mood disorders in adult populations using DSM-IV-TR diagnostic criteria [26]. Interviewers used a timeline to plot social, occupational, and medical events throughout the period of the interview. Events on this timeline were used as memory cues for participants to identify the date of onset and offset of depressive symptoms and episodes. Dates of symptom onset and offset were captured at the level of month and year with default values placed on the first day of the month if more specific information was not available from the participant. The timeframe for baseline SCID interviews was birth to the date of interview (i.e., lifetime history). The SCID interview used a “decision tree” format to guide the interviewer through possible diagnoses as the interview proceeded [27].

Interviewers were graduate-level psychology students who had been trained to reliability by the study principal investigator (MdG) using practice interviews to reach consistent levels of accuracy in recognizing symptomology (κ = 0.90–1.00). Case conferences were used to reach group consensus among the study principal investigator and interviewers regarding diagnoses and reduce interviewer drift over the course of the study.

*Physical Examination and Medical History.* Past medical history data were obtained via interview by the study staff, including the following: diabetes type and duration, height, weight, prescribed diabetes treatment regimen, medical contraindications for participation, medical diagnoses, and number and severity of diabetes complications.

*Glycated Hemoglobin* (*HbA_1C_*). Glycated hemoglobin (HbA_1C_) was measured using whole blood collection from venipuncture. Whole-blood samples were then analyzed at local hospital pathology laboratories. This analysis is a measurement of glycated fractions of HbA_1C_, which reflects the glucose level in blood over a 2- to 3-month time span. The reference range for HbA_1C_ samples was 25 mmol/mol (4.3%) to 39 mmol/mol (5.7%).

### 2.2. Data Coding

In order to assess the number and duration of psychiatric episodes, data from Program ACTIVE SCID booklets were reviewed and coded for each of the 176 participants who completed baseline assessment interviews. The following variables were coded for each participant:

Onset Date. The date on which the participant met full criteria for the DSM-IV-TR diagnosis of the disorder.

Offset Date. The date on which the participant no longer met full criteria for the depressive disorder. For episodes of MDD, this was the date that the participant no longer met criteria for five of the possible nine symptoms required for the diagnosis.

Date of Full Remission. The date on which the participant had been without core mood symptoms (depressed mood or anhedonia) for 2 months was considered the date of full remission.

Other Depressive Diagnoses. Onset and offset dates were recorded for all other depressive disorders, including dysthymia, depressive disorder not otherwise specified, adjustment disorder with depressed mood, and simple bereavement. Because these diagnoses do not incorporate a partial remission course specifier per the DSM-IV-TR criteria, the offset date was considered to be the date on which symptoms ended, per participant report.

Recurrence Periods. Recurrence periods were calculated as the difference between the date of full remission of the previous episode and the start date of the subsequent episode.

Other Mood Disorders. Remission statuses for all other the mood diagnoses were as defined by the DSM-IV-TR. Disorders assessed for remission statuses included the following: bipolar I disorder, alcohol abuse, alcohol dependence, non-alcoholic substance abuse, non-alcoholic substance dependence, panic disorder with/without agoraphobia, agoraphobia without a history of panic disorder, social phobia, specific phobia, obsessive–compulsive disorder, somatization disorder, post-traumatic stress disorder, generalized anxiety disorder, anorexia nervosa, bulimia nervosa, and binge eating disorder. Episodes were coded as “current,” “partial remission,” “full remission,” and “diagnosis never present”, consistent with DSM-IV-TR diagnostic criteria.

### 2.3. Statistical Analyses

Statistical Analysis Software (SAS) 9.4 was used for data analyses. Depressive episode duration was calculated by subtracting the episode start date from the date of full remission. Episode duration and remission periods were described by descriptive statistics. The primary outcome was the time to recurrence of MDD following the remission of the previous episode. Kaplan–Meier survival curves were used for the first, second, and third recurrences to describe the time to next episode at each observed event time point. Subjects were censored at the date of interview if they did not experience a specific event of interest. We used frailty models to conduct statistical comparisons of two survival curves in order to account the correlation among multiple recurrences within a subject. NLMIXED procedure, available in SAS 9.4, was used to fit frailty models. A similar approach was used to analyze the time to remission considering up to three episodes of any depressive disorder within a participant. Finally, Kaplan–Meier curves and Log Rank tests were used to determine whether having diabetes before or after the first episode of any depression disorder resulted in the same remission time or length of episode. Statistical significance was designated as *p* < 0.05%.

## 3. Results

Participant demographic characteristics are presented in Table 1. Study subjects were on average 56.5 years old, with 74.4% being female and 61.9% being white. Of those who reported their income (n = 145), 60.6% had an income of USD 40K or less. All participants met the criteria for T2DM. The average number of MDD episodes was 1.8, with an average MDD episode lasting 23.4 months (SD 31.9, range 0.5–231.3). When all depressive diagnoses were combined, the total lifetime exposure to depression was 43.1 months (SD 46.5, range 0.5–231.3).

### 3.1. Frequency and Characteristics of All Depressive Episodes

Table 2 presents the frequency rates of psychiatric disorders and the nature of depressive episodes across the spectrum of mood related depressive disorders. On average, the participants experienced their first episode at age of 40.4 (SD = 15.9) years and reported a lifetime prevalence rate of 2.1 (SD = 1.1) depressive episodes. Participants were exposed to an average of 61.1 (SD = 84.6) months of any depressive disorder (range: 1.0–666.4 months). Median episode duration of each depression diagnosis was 13.9 months (IQR = 5.6–33.0 months; range: 0.7–660.6 months).

### 3.2. Characteristics of Major Depressive Disorder

Table 2 provides a summary of the descriptive characteristics of MDD episodes of the total of 312 lifetime MDD episodes that were reported. On average, participants experienced an average of 1.8 (SD = 0.9 episodes, maximum = 5) MDD episodes over their lifetime with median length of episode lasting 13.9 months (IQR = 5.6–31.9 months). The median lifetime cumulative exposure duration to all MDD episodes combined was 33.0 months (IQR = 12.9–63.1 months).

### 3.3. Censored and Coded Episodes

In order to calculate a conservative estimate of length for these episodes, 116 of the 312 major depressive episodes were censored and were coded with an artificial full remission date (the date of the participant’s baseline SCID interview).

### 3.4. Characteristics of Other Depressive Disorders

In this study, other depressive disorders (ODDs) were defined as a DSM-IV-TR depressive diagnosis that did not meet the criteria for an MDD, which included depressed mood, adjustment disorder with depressed mood, simple bereavement, dysthymia, and depressive disorder not otherwise specified. A total of 40 ODD episodes were reported across all participants, with the median duration of these episodes lasting 12.0 months (IQR 3.5–52.0 months).

### 3.5. Remission Periods

Table 3 presents the analysis of the characteristics of the remission periods between depressive episodes. Participants were stratified on the basis of the number of depression-free and depressive episodes, with the depression-free period defined as the time from birth to the onset of the first depressive episode.

### 3.6. Depressive Episode Remission

Figure 1a,b show the survival curves for the time in months to the first, second, and third remission of an MDD. The remission time from having the first episode of an MDD was significantly shorter than the remission time from having the second episode of an MDD (medians: 14.0 vs. 37.9 months, *p* < 0.0001). Of those who had at least three episodes, the second remission happened much sooner than the third remission (medians: 13.0 vs. 28.0 months, *p* = 0.006).

### 3.7. Recurrence of Depressive Episodes

The Kaplan–Meier survival curves illustrating the length of first, second, and third recurrence of MDD are depicted in Figure 1c,d. As evident in Figure 1c, the median time to the second recurrence was significantly shorter than the median length of first recurrence (80.6 vs. 138.0 months; *p* = 0.017). In comparison, Figure 1d shows no significant difference between the median duration of second and third recurrences (75.0 vs. 88.5 months; *p* = 0.375), a finding that may be due to the small sample size.

### 3.8. Episode Duration and Order of Depression vs. Diabetes Onset

Finally, to determine whether the duration of major depressive disorder (MDD) varied based on the order of the diagnosis of T2DM, we conducted a Kaplan–Meier survival analysis that was stratified by order of depression vs. diabetes onset (i.e., those whose first diagnosis was depression vs. those whose first diagnosis was T2DM). No significant difference was observed in the survival data between the two groups. Participants diagnosed with T2DM before remission from their first MDD experience had a median episode duration of 14.0 months compared to 13.9 months for those diagnosed with T2DM after MDD remission (*p* = 0.882).

## 4. Discussion

The current study investigated the episode length, remission duration, and episode progression during the lifetime course of MDD and other depressive disorders in a sample of adults with T2DM from rural and urban geographical regions. Additionally, we explored the duration of the MDD episode and remission and its occurrence before and after the diagnosis of T2DM to determine whether the order of diagnosis influenced the overall burden of depression.

The median duration of MDD episodes in this study was 13.9 months compared to the median duration of 22 weeks observed in the general population as reported by Solomon and his colleagues [28]. The results of this study are consistent with the findings of other researchers, who found that depressive disorders tend to be more prevalent among T2DM patients than in the general population [9,10]. A review of the studies regarding the episodic duration of depression in the general population across recurrences has shown mixed findings with some researchers reporting similar durations between episodes (inter-episode recovery) [28], while others have found a shorter time lapse with each subsequent episode [29,30]. The recovery period between episodes decreased for patients who had two or more depressive episodes. The first episode of depression had the longest duration, with a median of 14–25 months, while subsequent episodes lasted a median of 8–9 months. We observed that the mean episode duration of MDD in our study was an average of 13 months longer than the mean duration of episodes that were reported by Lustman and his colleagues [31] in a mixed sample of adults with type 1 and type 2 diabetes.

Prolonged exposure to depressive episodes increases the risk of poor glycemic control [12], functional disability [18], increased severity and number of diabetes-related health complications [13], poor adherence to diabetes self-management behaviors [14,15], reduced quality of life [16,17], and premature mortality [19]. The findings highlight the critical importance of depression screening and early assessment of a clinical diagnosis of MDD paired with medication and or evidence-based psychotherapeutic interventions to short-circuit the progressive path of depressive episodes. Given the prolonged nature of depressive episodes in adults with T2DM (i.e., 13.9 median episode duration), follow-up screening at each subsequent medical visit (every 3–6 months) following diagnosis and repeated evaluation of treatment efficacy would be indicated.

Similarly to our previous study [20], we did not find an association between the length of the depressive episodes and whether the diagnosis of MDD or T2DM occurred first. However, previous longitudinal studies [32,33] have found that after controlling for gender, obesity, and socio-economic variables, a lifetime history of depression increases the risk of the onset of T2DM by 38–60%. This suggests that irrespective of whether depression was diagnosed prior to or after the T2DM diagnosis, the comorbidity of T2DM with depression may prolong the depressive episode and the concomitant burden that accompanies a depressive diagnosis.

The findings of this study must be examined in the context of several limitations. The study was designed as a randomized controlled effectiveness trial of two behavioral interventions, exercise and CBT, for adults with T2DM and MDD and, therefore, excluded patients without diabetes. The course of MDD could not be compared between those who are diagnosed with T2DM and those without diabetes in the same cohort. It must be noted, however, that we utilized diagnostic assessment methods that are consistent with those used by researchers to diagnose and plot episode length in the general population, and our methodology has diagnostic validity to compare our findings with those of studies in the general population. Secondly, at screening, we excluded individuals with current suicidal ideations, bipolar disorder, or substance abuse disorders, which may have resulted in selection bias, decreased generalizability to the entire population of adults with T2DM, and shortened the length of the reported durations of depressive episodes. For example, approximately 25% of adults with an MDD have a comorbid substance abuse disorder and those with substance abuse disorders are 2.3 times more likely to have comorbid MDDs in the general population [34]. Including these disorders could result in overestimating episode rates or add misdiagnosed disorders to the sample. Despite this exclusion, even as a conservative estimate, the episode lengths were observed to be longer than those reported for the general population. Third, the retrospective design, which is similar to previous studies [28], to assess the period of birth to baseline to assess depression during the SCID interview is subject to recall error, which may have influenced the accuracy of the data. To address this limitation, interviewers utilized a timeline method to minimize the effects of this limitation. This approach is not a failsafe for recall bias. Fourth, the sample sizes in some diagnostic subgroups and among men were small, thereby limiting the generalizability of findings to all people with T2DM. Finally, lifetime depression treatment (medication or counseling) was not assessed. It was not possible to accurately assess the effects of antidepressant medications in this sample when compared with the general population.

## 5. Conclusions

In conclusion, this study documents that clinical depression is a recurrent disorder with episode durations that are an order of magnitude longer than those observed in the general population. Irrespective of whether depression is diagnosed prior to or after the diagnosis of T2DM, depression in these patients ought to be treated with evidence-based behavioral and medication regimens to reduce the adverse consequences associated with the prolonged exposure to depressive disorders that are comorbid with T2DM.

## Figures and Tables

**Figure 1 healthcare-13-01773-f001:**
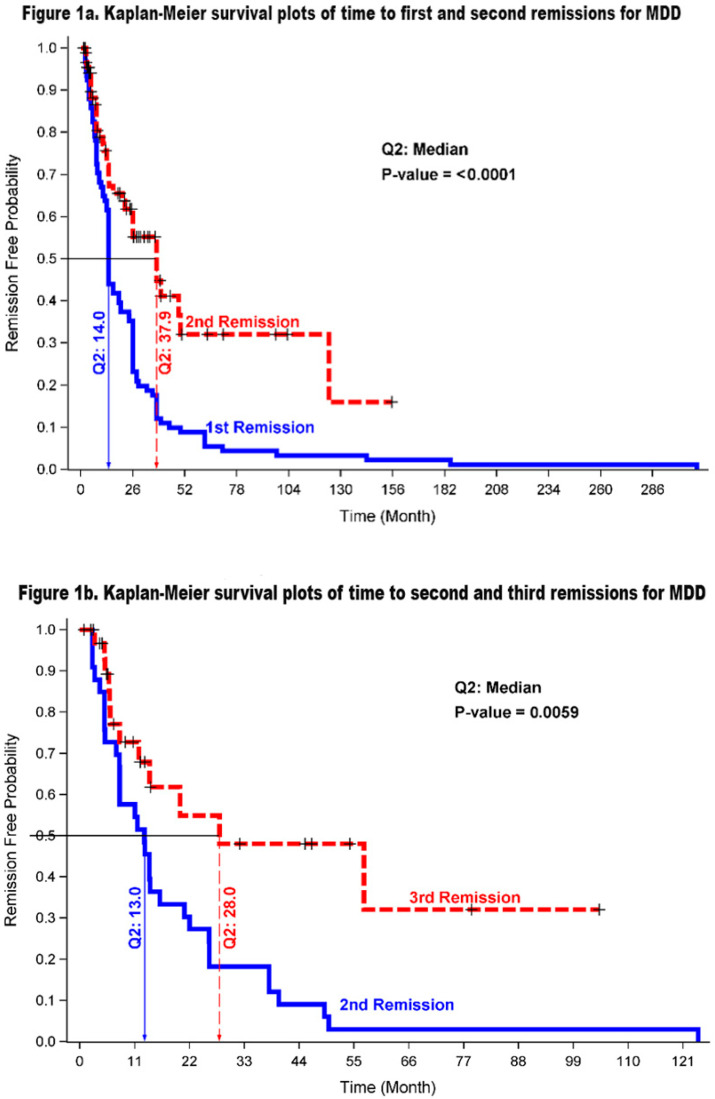
(**a**). N = 91 cases of first and second remissions from MDD. (**b**). N = 33 cases of second to third remissions from MDD. (**c**): N = 35 cases with first- and second-episode recurrences. (**d**). N = 12 cases with second- and third-episode recurrences.

**Table 1 healthcare-13-01773-t001:** Demographic characteristics of N = 176 Program ACTIVE II participants.

	Mean, N	SD, %
**Age**, *y* (mean, SD)	55.5	10.4
**Gender**		
Female	131	74.4
**Ethnicity**		
White	109	61.6
Black/African American	33	18.8
Other	4	2.2
Unknown or refused	30	17.0
**Education**		
HS diploma/GED or less	41	23.3
Trade school/part college	66	37.5
4-year college/post	61	34.7
Other	8	4.5
**Marital status**		
Married/living with partner	91	51.7
Not married	85	48.3
**Income**		
Missing	31	17.6
0–10,000	14	8.0
11,000–20,000	24	13.6
21,000–40,000	50	28.4
41,000–60,000	23	13.1
61,000+	34	19.3
**Health insurance (yes)**	130	73.9
**Current primary care provider (yes)**	139	79.0
**Current diabetes specialist (yes)**	35	19.9

**Table 2 healthcare-13-01773-t002:** Depressive episode characteristics.

	N	Mean/ Median	SD/IQR	Minimum	Maximum
**Number and Cumulative Exposure of Depressive Episodes**					
Age at First Onset of any Depressive Episode (years)	176	40.4	15.9	2.6	78.9
Mean Number of Depressive Episodes (MDD + ODD)	176	2.1	1.1	1.0	6.0
Mean Cumulative Exposure to All Depressive Disorders (months)	176	61.0	84.6	1.0	666.4
Median Cumulative Exposure to All Depressive Disorders (months)	176	37.5	15.3–70.4	1.0	666.4
Mean Duration of All Depression Episodes (months)	361	29.9	54.5	0.7	660.0
Median Duration of All Depression Episodes (months)	361	13.9	5.6–33.0	0.7	660.0
Mean Number of MDD Episodes	174	1.8	0.9	1.0	5.0
Mean Lifetime Cumulative Exposure Duration to MDD (months)	174	48.0	61.4	1.0	460.7
Median Lifetime Cumulative Exposure Duration to MDD (months)	174	33.0	12.9–63.1	1.0	460.7
Mean MDD Episode Duration (months)	312	26.7	39.0	0.8	308.0
Median MDD Episode Duration (months)	312	13.9	5.6–31.9	0.8	308.0
Mean Number of ODD Episodes	40	1.2	0.6	1.0	3.0
Mean Cumulative Exposure to ODD (months)	40	61.3	121.3	0.7	660.0
Median Cumulative Exposure to ODD (months)	40	12.0	3.5–52.0	0.7	660.0
Mean ODD Episodes Duration (months)	49	50.0	109.3	0.7	660.0
Median ODD Episode Duration (months)	49	12.0	3.5–36.0	0.7	660.0
**Episode Duration of MDD (Onset to Full Remission)**					
Mean Duration of Index Episode (Months)	166	29.0	40.2	0.8	297.2
Mean Duration of Earliest Pre-Index Episode (Months)	88	26.1	41.1	2.4	308.0
Mean Duration of Second Earliest Pre-Index Episode (Months)	32	19.8	23.4	2.5	124.0
Mean Duration of Third Earliest Pre-Index Episode (Months)	11	15.0	15.8	3.0	57.0

**Table 3 healthcare-13-01773-t003:** Well intervals following any depressive episode (years).

	1 DF Periods, N = 67 Mean (SD)	2 DF Periods, N = 57 Mean (SD)	3 DF Periods, N = 34 Mean (SD)	4 DF Periods, N = 12 Mean (SD)	5 DF Periods, N = 6 Mean (SD)
Between DOB and First Depressive Episode Onset	52.6 (10.9)	37.2 (13.7)	32.0 (11.8)	25.0 (10.1)	14.0 (8.5)
First Post-Depression Well Interval		12.7 (9.7)	11.0 (7.9)	8.2 (5.4)	5.5 (3.9)
Second Post-Depression Well Interval			8.1 (7.3)	6.9 (5.5)	6.6 (3.6)
Third Post-Depression Well Interval				7.8 (6.8)	4.8 (5.1)
Fourth Post-Depression Well Interval					2.7 (2.4)

Note. DF = depression-free, DOB = date of birth.

## Data Availability

Data is available upon request to the corresponding author.

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
