# Peer review of "A Post-Hoc Analysis of Depressive Disorders in Patients with Type 2 Diabetes"

_healthcare, 2025, doi:10.3390/healthcare13151773_

Round 1
Reviewer 1 Report
Comments and Suggestions for Authors
The study conducts a secondary analysis of Program ACTIVE II clinical trial data to investigate major depressive disorder lifetime trajectories in adults with T2DM. Through the use of the Structured Clinical Interview for DSM-IV-TR (SCID), the authors evaluated 176 participants to understand episodes duration and patterns of remission and recurrence of depression. The study reveals that depressive episodes last significantly longer in diabetic patients compared to the general population with median durations of 13.9 months versus the typical 3-6 months. The research question holds clinical relevance and uses sound methodology, but the conclusions require further attention because several concerns affect their strength before publication.
Required revisions:
- Correct Misrepresentation of Study Design
The title presents it as a "naturalistic observational study" while the manuscript clarifies it to be a secondary analysis of the Program ACTIVE II RCT (lines 102-104). The misrepresentation leads readers to misunderstand selection biases and generalizability. The title needs to be updated to correctly represent the study as "secondary analysis" or "post-hoc analysis."
- Clarify Statistical Significance Threshold
The methods state: The threshold for statistical significance was set at p < 0.5% (line 267). The statistical significance level of p < 0.005 (0.5%) diverges from standard practice in observational research studies. Determine if statistical significance should use p < 0.05 (5%) or provide citation support for the 0.005 threshold. Revise all statistical interpretations accordingly.
- Expand Discussion of Generalizability Limitations
The exclusion criteria you use for your participants (such as suicidal ideation, bipolar disorder, substance abuse and certain medical conditions in lines 114-118) yield a study sample that does not accurately represent actual T2DM populations. Although essential for the initial RCT design this approach greatly reduces external validity. The limitations section (lines 442-445) requires quantitative analysis of how these defined exclusions could lead to biased episode duration estimates. For example:
- What proportion of T2DM patients usually does not meet these research criteria?
- Are there underlying mechanisms where comorbid conditions such as substance abuse extend depressive episodes beyond what your dataset can measure?
- Remove Unsupported Claims About Antidepressant Efficacy
You state: The study asserts that antidepressants fail to treat depression (line 456) yet fails to provide data on how patients adhered to medication regimens, what doses they received, or how long the treatment lasted. The conclusion lacks methodological validity because it is based on insufficient treatment data. Either: Remove this speculative claim entirely, or incorporate a subsection that examines patterns of antidepressant use based on trial records.
- Address Recall Bias Robustness
The use of memory cues during line 153 cannot prevent substantial recall inaccuracy from retrospective SCID interviews about an entire lifetime span. Strengthen the limitations (lines 442-445) by:
- Examine how your data on episode duration compares with findings from prospective depression studies in T2DM patients.
- Run sensitivity analyses after removing participants who experienced more than three episodes due to reduced recall reliability.
- Standardize Terminology
ODD: The paper should introduce "Other Depressive Disorders" when it appears first (line 29) and must not change to "other mood disorders" later (line 195) unless it provides clarification.
Episode vs. Exposure: The Results section should explicitly differentiate between the median duration of individual episodes (13.9 months) and the median cumulative lifetime exposure (33 months).
- Add Confidence Intervals
The report should include 95% confidence intervals for median episode durations (e.g., "median 13.9 months [95% CI: X–Y]"). The skewed distribution of episode lengths which range from 0.7 to 660 months makes this critical.
- Enhance Figure Interpretability
Figures 1a–1d: Add numbers at risk below survival curves. For example:
"At 24 months: The count of individuals at risk during the first episode totals n=XX while the second episode has n=YY at risk.
Figure Legends: The study analysis should state whether it includes censored episodes totaling 166 instances.
- Verify Citations
- Line 459 cites Lustman et al. The paper mentions Lustman et al. but fails to indicate the specific type of diabetes studied (T1DM vs. T2DM). Clarify relevance.
- Line 435 cites Solomon et al. The citation of Solomon et al. must correspond to the "general population" comparison referenced.
- Explore Sex-Specific Analyses
Your study contains 74% female participants as indicated on line 272 but lacks reporting on sex-based differences concerning episode length. To address known differences in depression chronicity between sexes conduct separate sex-based analyses or control for sex in statistical models.
- Discuss Clinical Implications of Episode Duration
The 13.9-month median episode duration which exceeds the 3–6 month timeframe seen in the general population requires specific clinical commentary.
- What monitoring intervals should be implemented for T2DM patients who suffer from MDD according to this data?
- Does prolonged duration suggest treatment-resistant depression phenotypes?
Author Response
Please see attached document below.

Reviewer 2 Report
Comments and Suggestions for Authors
The main question addressed by the research focuses on describing the characteristics of depressive episodes and periods of remission in patients with T2DM. It also examines whether the timing of T2DM diagnosis has an impact on the duration or remission of depressive episodes.
The topic is both pertinent and original. According to the manuscript, the comorbidity of depression and type 2 diabetes mellitus (T2DM) is a well-documented problem that has a significant impact on glycemic control, diabetes complications, and quality of life. The study used standardized psychiatric interviews (SCID) to examine the lifetime course of major depressive disorder (MDD) in a T2DM population. This adds specificity to an area where previous studies have often relied on symptom-based tests or smaller sample sizes. Its relevance to different clinical settings is enhanced by including populations in urban and rural areas, as well as examining the duration of episodes and remissions before and after T2DM diagnosis.
Gap Addressed: By using a large sample and standardized diagnostic instruments, which were lacking in previous studies due to small sample sizes and homogeneous groups, the manuscript addresses a specific gap. Additionally, it examines the effects of diagnostic order (depression versus DZT2), an issue that has not been sufficiently studied. However, no significant differences were found. Furthermore, the focus on episode duration compared to the general population highlights a gap in understanding how this population experiences depression over the long term.
A robust statistical approach, using frailty models and Kaplan-Meier survival curves to examine the durations of recurrence and remission, showed that subsequent episodes of major depressive disorder (MDD) are longer (median second episode 37.9 months vs. 14.0 months for first episode, p=0.0001), and remission periods are shorter
The study investigates whether the sequence in which depression and type 2 diabetes mellitus (T2DM) are diagnosed influences the length of episodes, but it finds no discernible difference. This is consistent with the authors' earlier research.
Given the impact of prolonged episodes on health outcomes, it emphasizes the significance of early diagnosis and care for MDD in individuals with type 2 diabetes.
Suggested Improvements:
- Exclusion Criteria: Dropping participants with suicidal thoughts might mess with the results, having an impact on the results for less severe depression..It’d be great if the authors included a sensitivity analysis or at least discussed how this choice compares to studies examining the general population.
- Antidepressant Use: The inability to assess the effects of antidepressants is mentioned in the paper. Adding some details—like what meds people were on, how long they took them, or if they stuck with it—would help figure out if long depressive episodes are due to meds not working or something else entirely.
- Statistical Clarity: On line 186, the authors mention the p-value cutoff is below 0.5%, which is a slip-up (probably meant 0.05). Fixing that would make the stats way easier to follow.
The conclusion is appropriate with the data by describing the lifetime course of MDD (episode length, remission duration, recurrence) and confirming no influence of diagnostic order.
The references are adequate, with a mix of current and fundamental literature supporting the study's context and methods. The tables and figures are informative but could be streamlined for clarity.
Author Response
Please see attached response to reviewers.

Round 2
Reviewer 1 Report
Comments and Suggestions for Authors
No more comments,
All previous points had been addressed by the authors.
Reviewer 2 Report
Comments and Suggestions for Authors
Accept in present form.